# ORFA: Exploring WebAssembly as a Turing Complete Query Language for Web APIs

## Abstract

Web APIs are the primary communication form for Web services, with RESTful design being the predominant paradigm. However, RESTful APIs are typically fixed once defined, causing data under- or over-fetching as they can't meet clients' varying Web service needs. While semantic enriched API query languages like GraphQL mitigates this problem, they still face expressiveness limitations for logical operations such as indirect queries and loop traversals. To address this, we propose ORFA (One Request For All), the first in literature that employs WebAssembly (Wasm) as a Web API query language to achieve complete expressiveness of client requests. ORFA's key advantage lies in its use of Wasm's Turing completeness to allow clients to compose arbitrary operations within a single request, thus significantly eliminating redundant data transmission and boosting communication efficiency. Technically, ORFA provides a runtime for executing Wasm query programs and incorporates new module splitting strategies and a caching mechanism customized for integrating Wasm into Web API services, which can enable lightweight code transfer and fast request responses. Experimental results on a realistic testbed and popular Web applications show that ORFA effectively reduces latency by 18.4% and network traffic by 24.5% on average, compared to the state-of-the-art GraphQL.

## Keywords

Web API, WebAssembly, Query Language, Expressiveness, Runtime

## 1 Introduction

In modern Web systems, Web APIs play a crucial role as the primary method of co-operation and communication between -services [1, 2, 3], particularly in microservice architectures [4, 5]. Web service interfaces are required to support increasingly complex network services and have evolved from traditional Restful APIs [6] to more flexible solutions such as GraphQL [7]. As illustrated in Figure 1, different clients may request various types of information through the API to interact with the Web server. Despite varying client needs, RESTful APIs are generally fixed in service, which can easily cause data over-fetching and under-fetching in practice [8, 9, 10, 11]. Over-fetching occurs when the server's response includes more data than the client requires, leading to unnecessary network transmission costs, while under-fetching happens when the data returned is insufficient, forcing the client to make additional requests.

GraphQL is the state-of-the-art query language that can mitigate these data over-fetching and under-fetching issues at Web services, given its enhanced expressiveness. For instance, in a scenario where client *A* only needs *create_time* and client *B* requires only the *last_login* time of the target user, both clients use a *GET /user/{id}* request if using RESTful style APIs. This causes unnecessary network

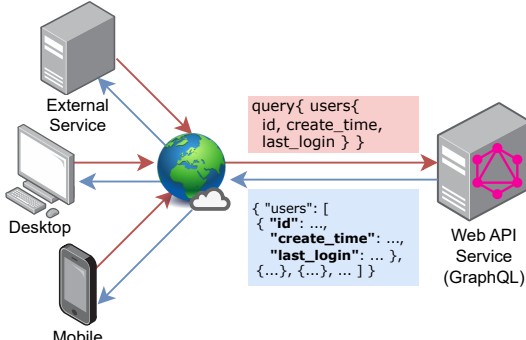

**Figure 1: A typical modern Web API service example that supports various clients with varying needs.**

transmission since redundant information will be returned using this query. In contrast, with improved expressiveness of GraphQL, the clients can submit *query{user{id, create_time}}* and *query{user{id, last_login}}* to acquire the exact required information, effectively saving network resources by eliminating over-fetching. On the other hand, under-fetching can also occur by using RESTful style, which causes significant back-and-forth communications across the network. As shown in Figure 2 (top), *GET /user/{id}* request is for acquiring detailed information of user *{id}* like privilege roles; and *POST /user/{id}/notice* for sending notification. Then, to implement the logic of "*sending notifications to admins*", the client needs to send $1 + m + n$ requests, where $m$ is the user count and $n$ is the admin count. In contrast, the same operation can be accomplished via GraphQL in just two requests, as detailed in Figure 2 (middle). Apparently, enhanced expressiveness significantly helps shorten the operation time and reduces the data transmitted.

**Expressiveness limitations of GraphQL.** Despite its improved expressiveness, GraphQL still has a critical limitation in that *it is not Turing complete*, meaning not all operations can be accomplished within a single request. In practice, some common logic patterns such as indirect queries [12] and loop traversals [13] remain inexpressible by GraphQL. For instance, due to GraphQL's inability to mix *query*s and *mutation*s, at least two requests are needed to accomplish the task, shown in Figure 2 (middle). If we elevate the expressiveness of the query language to a Turing complete level, e.g., allowing clients to send a program as a query, then it theoretically enables arbitrarily complex operations performed in a single query, realizing the full potential of **"One Request For All"**. Demonstrated by Figure 2 (bottom), with a Turing complete query language (like C), only **one** request is needed for this task. Thus, there still remains significant potential for further enhancing expressiveness.

However, implementing this Turing complete idea poses practical challenges. Executing client-provided programs on the server

*WWW '25, April 28–May 02, 2025, Sydney, Australia*
2024. ACM ISBN 978-x-xxxx-xxxx-x/YY/MM
https://doi.org/10.1145/nnnnnnn.nnnnnnn

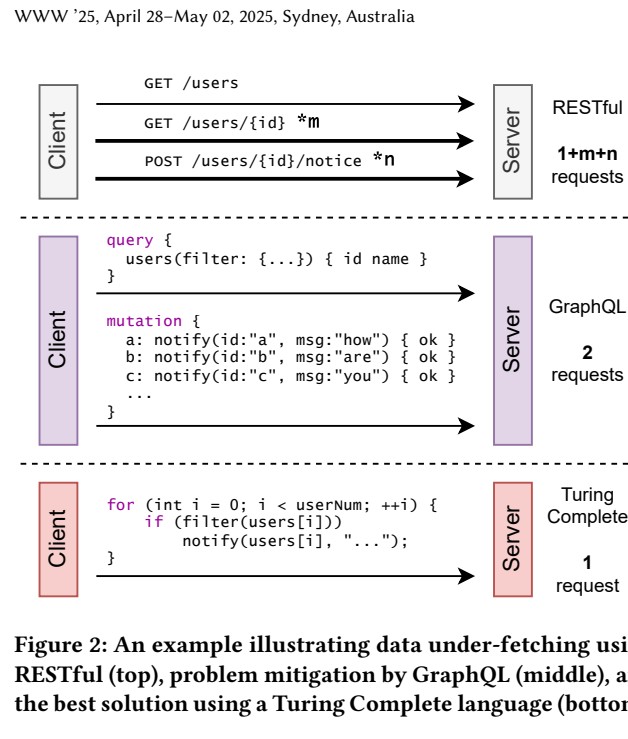

**Figure 2: An example illustrating data under-fetching using RESTful (top), problem mitigation by GraphQL (middle), and the best solution using a Turing Complete language (bottom).**

introduces data and resource security risks, as these programs might exploit vulnerabilities or overconsume resources. To address these risks, the programming language used for queries and its interpreter must enforce strong data isolation and resource constraints, which traditional languages often lack. Fortunately, WebAssembly (Wasm) [14] meets the strict security requirements. Wasm is a Turing complete intermediate representation (IR) with built-in performance and security mechanisms, originally for running server-sent programs in client browsers with strong safety guarantees. Moreover, the core of Wasm [15] is neural and general-purpose, making it suitable for applications beyond the browser.

Building on these insights, this paper explores the novel use of Wasm as a Turing-complete query language for Web APIs. Traditionally, Wasm is employed in a server-to-client model, where the server sends Wasm binaries to the client (often a Web browser) for secure execution in a sandbox environment. Our approach reverses this conventional flow by enabling clients to send queries as Wasm programs to the server, which poses unique implementation challenges. ① The foremost problem is the programming model, i.e., how should the Wasm program be written, executed, and debugged in such a new querying scenario? ② Although Wasm is more compact than traditional binary programs like x86 ELF files, it is still too large for most query use cases. Typical queries are only a few kilobytes in size, whereas even the simplest *hello-world* Wasm program can exceed 100 kilobytes, which can greatly burden the request initiation. ③ Unlike GraphQL, Wasm programs spend much more time on compilation and instantiation before execution, thus necessitating an effective solution to reuse previously served programs, particularly for repeated queries.

We give our solutions to the above-mentioned issues in this paper. Specifically, the contributions of this work are as follows:

(1) We highlight the necessity of enhancing expressiveness for Web API requests and the imperfection of the SOTA GraphQL in terms of completeness, which motivate us to propose ORFA, a Web-oriented framework employing Wasm as the query language to achieve Turing completeness and reaches the goal of **"One Request For All"**.

(2) We introduce ORFA's programming model and explain how to program, execute and debug the query programs. To reduce the size of the query module, we propose a novel module splitting technique that utilizes Wasm's inherit *import/export* functionality and avoids relocation overhead in existing linking methods. We also design a caching mechanism for ORFA that significantly reduces the startup latency, program transmission, and resource usage at the servers. Our mechanism achieves a new application of Wasm to Web API querying scenarios with effective solutions addressing the program size and startup problems simultaneously.

(3) Evaluations on representative system and workloads demonstrate that ORFA remarkably reduces request latency and network traffic, effectively outperforming the traditional RESTful APIs and the state-of-the-art GraphQL.

## 2 Background and Related Works

**RESTful Web API.** Modern Web systems rely on Web APIs for inter-service communication and co-operation [1, 2, 3], especially in distributed and microservice architecture [4, 5]. Although many protocols can be used for Web APIs, such as SOAP [16], JSON-RPC [17], etc., RESTful style that directly utilizes the elements in the HTTP protocol has become the default choice for Web API design [18, 19, 3, 5]. REST [6] is not a specific protocol, but rather a vague set of design rules and guidelines. OpenAPI [20] specification is an effort to formalize and standardize REST that defines a format to describe and document APIs in an organized and predictable manner, serving both for humans and machines.

**GraphQL and Query Languages.** GraphQL has become a popular supplement and alternative for traditional RESTful API design. By 2023, as many as 23% software projects have adopted GraphQL [21], with industrial companies such as GitHub, Shopify, and Yelp implementing it. Practical evidence has demonstrated that GraphQL can significantly reduce engineering efforts, accelerate development [9], and decrease communication overheads [8], strongly demonstrating the necessity and feasibility of enhancing expressiveness. Netflix previously addressed the inflexibility of traditional Web APIs with Falcor [22], a JavaScript library rather than an formal query language. However, due to GraphQL's growing popularity, Netflix has discontinued Falcor. Other techniques, such as OData [23] and HT-SQL [24], embed SQL queries into HTTP URLs for client request customization. But they are limited to specific application scenarios and thus do not generalize well for broader Web API use cases. Query languages are more commonly associated with database systems, as seen with graph query languages including SPARQL [25], Cypher [26], Gremlin [27], and more. These database scenarios are different from Web services in that databases manage well-structured, static data, while Web services handle more dynamic and client-specific interactions. Therefore, Turing completeness is not the focus and primary goal of database works.

**WebAssembly (Wasm)** was proposed to address the performance limitations of JavaScript on the current Web platform [14]. The strict

security limitations of browsers ensure that Wasm is executed in isolated sandbox environments with strong safety features. Although originally designed for the Web, Wasm's language design avoids introducing Web-specific components and keeps its core purely computational. As a result, it has become an ideal, general-purpose intermediate representation suitable for various systems and has been widely applied to many outside-browser domains, including cloud and serverless computing [28, 29], high-performance computing [30, 31], and the Internet of Things [32, 33]. Similarly, Wasm can be very promising to reshape and empower the query-based Web systems.

**Wasm-based Server-side Remote Execution.** The native use of Wasm is to execute programs sent by Web servers, inside client browsers. But this paper aims at the reverse, i.e., sending query programs from the clients to be executed on the server. In fact, the practice of sending Wasm to servers for remote execution is not new, with one common usage to offload computation from clients to servers for both Wasm [34, 35, 32] and JavaScript [36, 37, 38]. Nonetheless, the Web API querying scenario focused by ORFA differs significantly from these works in terms of the program size, execution time, and job amount by orders of magnitude. Therefore, these techniques cannot replace ORFA. Wasm has also been widely explored for serverless systems as a lightweight alternative for Linux containers [39, 40, 29], where the Wasm programs act as remote executions from the perspectives of serverless developers. The Wasm programs in these systems function as normal Web services and are pre-uploaded to the serverless platform, whereas ORFA's query programs are dynamic and unpredictable. As far as we know, no existing works have used Wasm as a query language for Web APIs to enable the complete expressiveness of client requests.

## 3 ORFA

### 3.1 Overview

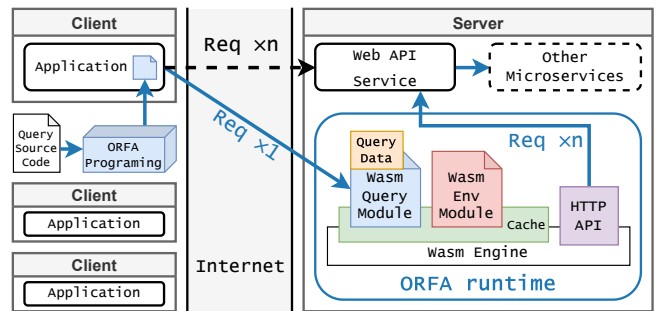

**Figure 3: ORFA is a Wasm-based framework consisting of programming supports and server-side runtime. The Wasm query modules are programmed into client applications and then sent to remote runtime for execution to issue multiple local requests in substitution of original remote requests.**

Figure 3 illustrates the overall architecture of ORFA, a framework consisting of programming supports and server-side runtime. The programming support aims to assist client developers to compose their consecutive Web API operations into a Wasm query module, embedded as part of the client application. The runtime is deployed

as a microservice alongside existing Web API microservices, minimizing communication costs between them. At run time, the client application sends the query module along with the associated query data to the ORFA runtime for remote execution. The query module is then combined with the environment module preloaded on the server and executed within the Wasm engine, with existing query program instances being reused if the cache hits. During execution, the query module can perform arbitrary computations and send requests to system-specified Web API services via ORFA's HTTP APIs. The Turing completeness of Wasm ensures that any complex operational logic can be encapsulated within a single query module. This approach allows the original **n** cross-internet remote Web API requests to be reduced to **1** remote request plus **n** inexpensive local requests, thereby reducing overall operational latency and network traffic. On the other hand, for Web API services, since ORFA enables users' customization for query operations, service developers now can refine Web API granularity to eliminate redundant functionalities, thus reducing service code maintenance costs.

**Table 1: Additional headers defined by ORFA.**

| Header | Note |
|---|---|
| ORFA-Input | Specify the length of input data in the message body, used to separate input data and Wasm module code. |
| ORFA-Limit | Specify the required time and space for program execution. |
| ORFA-Debug | Used in debugging mode. |
| ORFA-Cache | Specify cache mode and cache token for caching mechanisms. |
| ORFA-Trust | ECDSA signature of Wasm module, used for verifying the integrity of the received program. |

The clients communicate with ORFA via the HTTP protocol, with additional headers supporting ORFA's functionalities, as summarized in Table 1. The main components of the client's request are the query data and the Wasm query module, shown in Figure 3, which are encoded in binary and concatenated to form the HTTP request body, with the *ORFA-Input* header indicating the boundary in between. If the query succeeds, ORFA puts the result in the HTTP response body and returns it to the client. The specific content and encoding of the response body are entirely determined by the query module. Here are two major differences between ORFA and GraphQL: First, ORFA enforces the separation of query data and the query program, whereas GraphQL allows the mixture of the variable parts and the query code (refer to Figure 2), though it does recommend the usage of *variables* to achieve such separation [41]. The enforcement caters to the usual static compilation usage of Wasm and plays a key role in supporting ORFA's caching mechanism (§3.4). Second, GraphQL defines its response body as JSON format corresponding to the request's query structure, while ORFA allows the query itself fully determines the response body, allowing autonomous selection of the most efficient and compact encoding method. This is attributed to the Turing complete expressiveness, which enables the computation required for encoding.

To prevent clients from abusing the computational resource, ORFA uses the *ORFA-Limit* header to constrain the resource usage during execution. The resource safety risks associated with enhanced expressiveness is unavoidable and also exists in non-Turing-complete GraphQL [42, 43, 44, 45], but the success of GraphQL demonstrates that these risks can be accepted in practice[1].

## 3.2 Programming Web API Queries

Unlike usual Wasm programs, ORFA defines the query's entry point as a Wasm function named "orfa", which accepts two i32 parameters representing the starting address and length of the query data in the request body, respectively. Since Wasm is a general-purpose intermediate representation supported by many programming languages, any language capable of generating such a Wasm function can be used to write ORFA's query modules. For simplicity and due to the maturity of the toolchain, we choose C as the source language in this work.

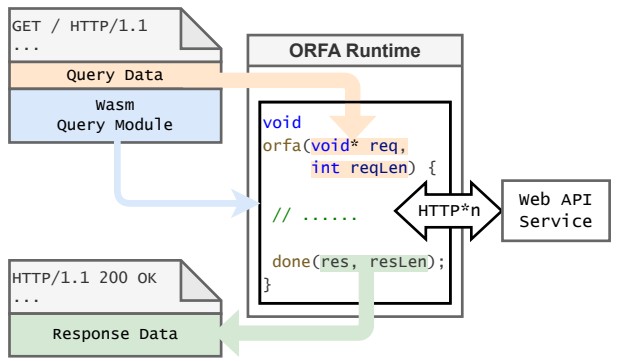

**Figure 4: Programming an ORFA query. The query data carried in the request body is passed as parameters. The query should submit the data to response body by calling done(). Arbitrary HTTP requests can be made to the specified Web API service.**

As shown in Figure 4, the entry point of the ORFA query program corresponds to a C function with the signature void *orfa(void*, int)*. Before executing the *orfa* function, ORFA places the query data from the request body into the Wasm program's address space and passes the starting address and data length as arguments. The query then executes from the beginning of *orfa*, where programmers can write arbitrary code for computation or calling functions from Table 2 to interact with external services. Finally, at the end of the query execution, the programmer should collect the necessary data and encode it into a contiguous address space, then call the *done* function to submit the data to the ORFA runtime. The data will be put into the response body and returned to the client by ORFA, thereby completing the query.

One thing to note about Table 2's API design is that the function used for issuing HTTP requests is asynchronous, thus allowing for the overlapping of multiple Web API operations. Also, it is important to point out that writing a practical query program requires

---

[1]Details of the solution to the resource safety risks, e.g. resource limiting policies, are omitted due to space limit.

**Table 2: Built-in functions in ORFA environment.**

| Function | Note |
|---|---|
| void done(const void*, uint32_t) | Submit the response data. |
| void Handle_del(Handle) | Delete an object. |
| int32_t Future_ready(Handle) | Check that whether a future object is ready. |
| Handle http(Request*) | Send a HTTP reqeust asynchronously, returning a future object handle. |
| int32_t Response_get(Handle, Response*) | Extract data from a future object if it's ready. |

significantly more supports than what is provided by the Wasm built-in instructions and Table 2 's APIs, such as dynamic memory allocation, string manipulation, JSON parsing, and more. Without these supports, writing a query program would be exceptionally difficult. However, these supports are essentially purely computational and can be implemented as Wasm functions. ORFA consolidates these basic supports into a common Wasm environment module for shared use across all requests. The specific implementation details will be discussed in §3.3.

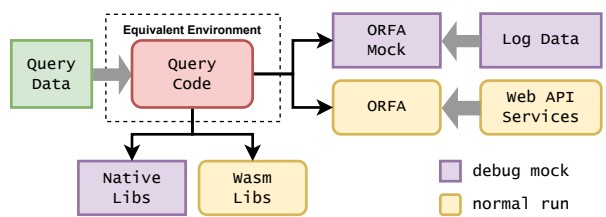

**Figure 5: The record and replay debugging of ORFA. The ORFA Mock tool together with native libraries ensures the equivalence of query execution environment.**

The complexity of query code greatly surges along with the enriched expressiveness, which thus crucially necessitates the support for debugging to facilitate query programming. In such Web API querying scenario, connecting a debugger to a remote Wasm runtime service is not feasible, as the queries are very short-live and the server needs to handle massive queries, hence unavailable for interactive debugging with programmers. Therefore, we choose an alternative design: recording and replaying the query program's execution. The availability of this approach highly relies on the deterministic property of ORFA's programming model: As the Wasm core is fully sandboxed and purely computational, by recording all inputs during the execution, the entire running process can be reproduced elsewhere. To enable the recording, clients set the *ORFA-Debug* header in its requests, and ORFA will respond the recorded log data instead of the original query result. The log data can be later used by our *ORFA Mock* tools at the client side locally, as shown in Figure 5. The query code links to different environment libraries in normal execution and debugging simulation,

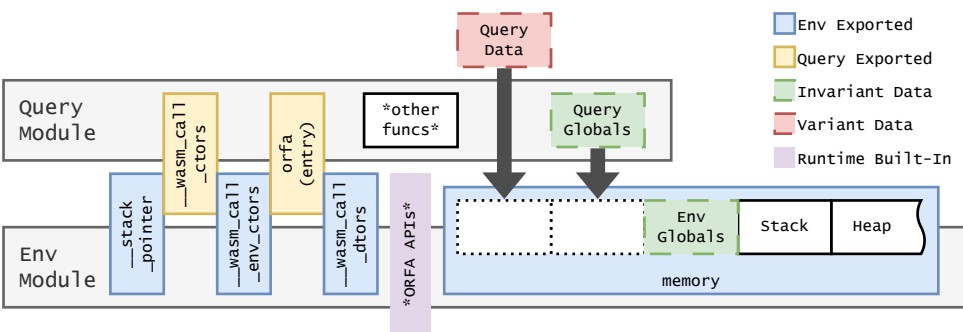

**Figure 6: The splitting and recombination of the query module and environment module. This figure shows the** *export*/*import* **relation of key elements of query and environment modules and the partition of memory space for the two modules.**

and *ORFA Mock* ensures the equivalence of the simulated environment between real remote environment. This way, the query code can be debugged locally like a normal program.

### 3.3 Shrinking Query Module Size

Using a normal Wasm module as the query program may bring an serious issue that the program size itself greatly outweighs the truly critical query data, thus potentially nullifying the traffic reducing benefits of composing multiple requests into one. This isn't to say that Wasm format is bloated; in fact, quite the opposite is true that Wasm programs are significantly smaller than typical binary programs (such as x86 machine code). However, in our query scenario, the programs are extremely small, with just a few kilobytes usually, whereas even the simplest Wasm "Hello, World!" program can exceed 100 KB. This discrepancy forces us to devise a method to reduce the size of the query module.

We have identified that this issue arises from the semantic gap between queries and Wasm. In detail, to support "simple" data extraction and assembly operations in queries, Wasm programs require a substantial amount of basic support code like dynamic string concatenation from standard libraries. But just including the *musl libc* from WASI-SDK [46] can cost over 1 MB, without consideration of other libraries. If we could separate these common basic codes from the query code and pre-load them onto the servers, it would eliminate the need to repeatedly transfer them over the Internet. Therefore, we propose a method to split a complete query Wasm program into a query module and an environment module. The environment module contains the common basic code, provided by the server and pre-loaded into ORFA. The query module includes query-specific variable code, provided by the client and combined with the environment module to form a whole Wasm program for execution.

Figure 6 explains the splitting and recombination of the two modules. A compiled Wasm program mainly consists of *function*s, which can be easily split and recombined using the *import* and *export* mechanisms. However, making functions from two separate modules work together requires additional conventions that are not explicitly defined in the Wasm specifications. A representative example is that both modules must agree on the memory layouts, which is reflected in the addresses used by all memory access instructions across all functions. Our design involves splitting the global variable segment in the memory space into two regions with one for

the query module and another for the environment. We then customize the Wasm linker to allocate different regions for the global variables of each module, ensuring that they do not overlap. To correctly access stack data, we export the global `__stack_pointer` from the environment module and import it in the query module, making both modules share a common stack. Finally, during the query initialization process, the global constructors generated in both modules, `__wasm_call_env_ctors` and `__wasm_call_ctors`, should be both invoked in order to ensure proper execution of the query code.

Be noted that our module splitting and recombination method is neither existent static linking [47] nor dynamic linking [48]. It is directly based on the Wasm's *import* and *export* mechanisms instead and requires no additional compilation information. Accordingly, one advantage of this splitting design is that it avoids the traditional linking overhead of redirecting all memory access instructions, and allows the environment module to be pre-loaded into ORFA.

### 3.4 Reducing Query Startup Latency

The execution of a Wasm query program consists of three phases, i.e., compilation, instantiation and execution. The first two phases are newly introduced compared to GraphQL. Considering that query requests are typically short-lived and massive in scale, the two phases may incur considerable startup overhead and latency. On the other hand, unlike GraphQL, whose programs are in text format and easy to be assembled dynamically, Wasm programs are in binary format and usually compiled statically from hand-written source code in high-level languages. As a result, Wasm query modules tend to remain unchanged during the run time of the client applications, leading to repeated compilation and instantiation of same query programs. Based on this observation, we extend ORFA with a caching mechanism to store the compiled results (referred to as the *ORFA-code* mode) or the initialized instances (referred to as the *ORFA-inst* mode). The *ORFA-inst* mode can help eliminate the startup overhead, achieving performance comparable to native code, but requires additional server memory and more careful coding of the query module to make the Wasm instance stateless and reusable.

To enable caching, the client must set the *ORFA-Cache* header in the request, specifying the desired caching mode and the previously cached token (if any). If the caching succeeds, the server then returns the refreshed cache token, and the client can omit the Wasm

query module part in later requests to further reduce the network overhead. For cache management, we employ a function-based [49] strategy, which exploits a background thread to periodically check the cache. If the number of cache items reaches a predefined threshold, the thread removes the least valuable cache items. The value of a cache item is calculated simply as the ratio of use counts to the time since last use.

It is common to see the same query programs sent from different front-end clients, as a client application usually serves many end-users. To share caches of query programs between clients, the value of the *ORFA-Trust* header is used as a key to retrieve the existing cache token at the first caching request. The *ORFA-Trust* value is a cryptographic signature of the query program, whose private key is generated by the client developers, and the corresponding public key is given to server maintainers and preset into the ORFA service. Using cryptographic signatures instead of plain hashes can also helps to avoid the risks of caching efficiency downgrading when malicious attackers flood the server with useless cache requests to exhaust the cache capacity. Such signature-based caching mechanism is made possible by the data-query separation design in ORFA's requests, and is not feasible in GraphQL for those dynamically assembled queries from clients, as it is impossible to sign the query program in advance.

## 4 Evaluation

To demonstrate the effectiveness of the proposed ORFA, we evaluate primarily from the following aspects. 1) **Efficiency** (§4.2): we apply ORFA in three widely-used realistic applications, and compare the latency and traffic metrics with those of GraphQL and the REST API; 2) **Sensitivity** (§4.3): we further conduct sensitivity studies to investigate the impact of network conditions and workflow complexities by adjusting client locations and the task workflow; 3) **Cost** (§4.4): to understand ORFA's service cost and its impact on other Web API services when sharing server resources, we collect the peak throughput of GraphQL and ORFA by stress testing with synthesized and realistic workloads respectively.

### 4.1 Experimental Methodology

**Table 3: Configurations of the experimental machines.**

|  | AWS t2.micro (Client) | Azure Standard B1s (Server) |
|---|---|---|
| RAM | 1G (+ 1G SWAP) | 1G (+ 1G SWAP) |
| CPU | Intel Xeon E5-2676 v3 @ 2.40GHz (1 vCPU) | Intel Xeon E5-2673 v4 @ 2.30GHz (1 vCPU) |
| OS | Ubuntu 22.04 | Ubuntu 22.04 |

*4.1.1 Node Testbed.* To model the real scenarios, our experiments are conducted on two virtual machines of different public cloud services: an AWS t2.micro and an Azure Standard B1s. These machines are designated to operate as the client and server, respectively. As detailed in Table 3, they roughly have equivalent configurations. And, for ORFA, we choose the popular outside-browser embedder, Wasmtime[2], as the Wasm engine.

*4.1.2 Workloads.* Three representative and popular applications on GitHub are chosen as benchmarks:

- **Gitea**[3] **(39k stars)**: a popular open-source Git server written in Go. Gitea only provides REST APIs specified with OpenAPI.
- **Memos**[4] **(21k stars)**: a self-hosted lightweight online note-taking service developed in Go. Similarly, it only provides OpenAPI-specified REST interfaces for third-party integration.
- **Strapi**[5] **(58k stars)**: a leading open-source headless content management system (CMS) developed purely in JavaScript. Strapi uses REST APIs as default, and also provides a GraphQL interface as a plugin.

For each application, we compose two types of workflows, with one for read-only query and the other for write-operation query (with data modification). In total, as listed in Table 4, there are six workflows, which are denoted with suffix *.r/.w*. For instance, the read-only workflow of *Memos* is *Memos.r*. Each workflow is further associated with a variable $N$, representing the complexity of the workflow. Notably, since *Gitea* and *Memos* only provide REST interfaces in OpenAPI format, we use the OpenAPI-to-GraphQL [50] tool to generate GraphQL wrappers.

*4.1.3 Metrics.* We focus on three common metrics, latency, network traffic, and throughput, to quantify ORFA's efficiency. The latency represents the time taken to complete the entire workflow, the network traffic is the amount of data transmitted during the workflow execution, and the throughput is the request number processed within a fixed time interval. The results are obtained using JMeter[6], a popular load testing tool.

*4.1.4 Comparison Designs.* We compare baseline and our proposed methods as listed below:

- **REST** represents that the operations are done by invoking REST APIs directly. The number of remote requests to complete each workflow is $N$.
- **GraphQL** depicts that the same operations are performed indirectly by GraphQL queries. Using GraphQL, these workflows need to first retrieve a JSON list, followed by batching operations on the elements in the list. Thus, the number of remote requests is always 2, regardless of the value of $N$.
- **ORFA-base** represents that the same operations are performed indirectly by sending a Wasm query module to ORFA without caching. Due to the improved expressiveness, the number of remote requests is always 1.
- **ORFA-code** is same as ORFA-base except that the compiled results are cached so that the code of the query module will not be sent repetitively. In this mode, the compilation is eliminated but the initialization is still required.
- **ORFA-inst** is same as ORFA-code except that the final instance is cached and reused too, so that all overheads of compilation and initialization are eliminated. ORFA-inst is our default configuration in continuously running serving processes.

---

[2]https://wasmtime.dev/
[3]https://github.com/go-gitea/gitea
[4]https://github.com/usememos/memos
[5]https://github.com/strapi/strapi
[6]https://jmeter.apache.org

**Table 4: Composed workflows of real applications.**

| Application | Read-only Workflow (.r) | Read-write Workflow (.w) |
|---|---|---|
| Memos | Get the second newest notes of each user. $N$ is the user number. | Change the visibilities of all notes from a user. $N$ is the notes number. |
| Strapi | Get related entries in table A for each entry in table B. $N$ is the number of entries in table B. | Add relations between entries in table A and a entry in table B. $N$ is the number of entries in table A. |
| Gitea | Get the second newest commits of all branches in a repository. $N$ is the number of branches. | Delete users whose name stars with a prefix. $N$ is the number of filtered users. |

## 4.2 Service Latency and Network Traffic

In this section, we place our Azure server in Singapore and the AWS client in Sydney. This setup leads to a communication latency of 93ms in between. Figure 7 presents the latency, and network traffic results of the six workflows, specifically when variable $N$ is set to a typical value of 4.

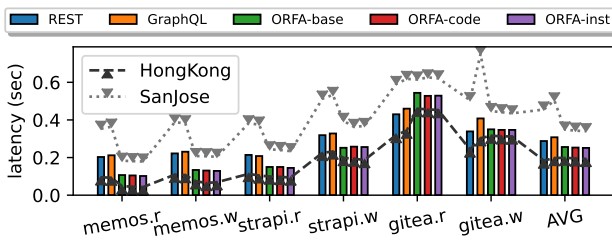

**(a) Latency results. The bars show the metrics measured in Sydney, while the metrics from Hong Kong and San Jose are depicted as grey and black lines (to be analyzed in §4.3).**

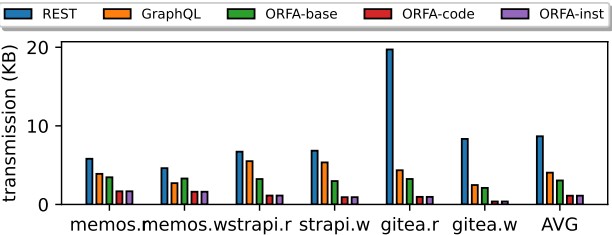

**(b) Network traffic results. This figure shows the total transmission amount for each workflow.**

**Figure 7: Experimental results of realistic applications.**

*4.2.1 Latency.* For the latency analysis, we execute each workflow ten times and choose the median as the final result, as shown in Figure 7a. ORFA poses lower latency across all work modes when compared to both REST and GraphQL, with the only exception of the *gitea.r* workflow. Particularly, ORFA-inst reduces at most 52% latency on *memos.r* compared to GraphQL, with an average of 18.4% reduction. In terms of *gitea.r*, the obviously long bar in Figure 7b and the observed latency issues are attributed to the parsing of the uncommonly long JSON data returned by Gitea's REST interface. In ORFA, this parsing process is conducted using cJSON within the Wasm interpreter, which is significantly less efficient compared to GraphQL's approach. GraphQL utilizes highly optimized JavaScript

engine code for parsing, leading to better performance in this workflow. This additional parsing overhead in ORFA becomes the primary contributor to latency, overshadowing the benefits gained from reduced network communication.

*4.2.2 Network Traffic.* Since the volume of data transmission is solely determined by the task and method, it remains consistent and is thus unaffected by variations in network conditions. Figure 7b presents the network traffic for each workflow. We can see that ORFA mostly has the least transmission volume, especially in the caching modes. On average, 24.5% traffic is reduced in ORFA-base and 72.4% in ORFA-code/ORFA-inst. This is because the increased expressiveness reduces the number of requests and allows the workflow-specific data encoding. There is only one exception: ORFA-base in the *memos.w* workflow. In this case, the data transmitted is not so much that the extra size of the Wasm query module diminishes the benefits of reducing one data round trip compared to GraphQL. As a result, ORFA-base's total transmission is slightly higher than GraphQL's.

Putting together, the latency and traffic results demonstrate that ORFA can effectively improve latency and network transmission, outperforming the existing REST and GraphQL.

## 4.3 Impacts of Network and Workflow

To further validate the efficiency and robustness of ORFA, we continue investigating the influence of network conditions and workflow complexities. Previously, we position the client in Sydney, with a delay of 93ms to the server, and choose a moderate value for $N$, being set as 4. For network conditions, we relocate clients to another two positions, Hong Kong and San Jose, and then observe changes in overall latency. For workflow complexities, we sweep over different values of $N$ to understand their effects on latency and network traffic.

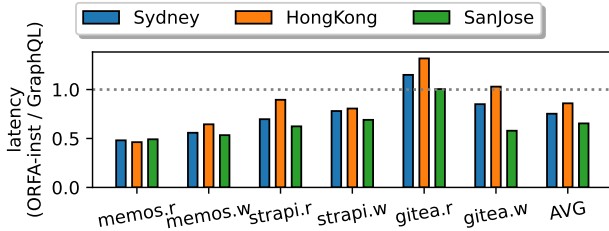

**Figure 8: The latency ratio of ORFA-inst to GraphQL when the client is in different positions (Lower is better).**

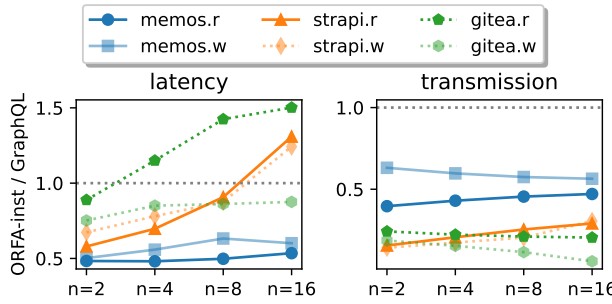

Figure 9: The latency and network traffic ratio of `ORFA-inst` to `GraphQL` when *N* varies (Lower is better).

*4.3.1 Network Conditions.* In terms of network conditions, clients in Sydney, Hong Kong, and San Jose respectively have a delay of 93ms, 35ms and 170ms. Since the data transmission amount is unaffected by network latency, we only analyze latency results. At first, we notice that the network condition change causes a shift on latency and the trend largely remains stable, which is illustrated in Figure 7a (§4.2). We further calculate the ratio between the latency of `ORFA-inst` and that of `GraphQL`, with results being reported in Figure 8. It can be observed that in almost all cases, `ORFA` achieves lower latency than `GraphQL` regardless of network conditions. Overall, `ORFA` can generally perform better with constrained network conditions, i.e., higher transmission delays.

*4.3.2 Workflow Complexity.* Regarding workflow complexity, we place the client in Sydney and choose 2, 4, 8, 16 for *N*. Figure 9 summarizes the latency and network traffic ratios between `ORFA-inst` and `GraphQL`. Overall, in most cases, `ORFA` still maintains advantages in all values of *N*, demonstrating the robustness of our design. In terms of the network traffic, `ORFA` consistently achieves lower network transmission. Besides, for latency and throughput, we observe that applications react differently to changes in the value of *N* and two workflows of the same application react similarly. *Memos*'s workflows are not very sensitive to *N*, as *Memos* is lightweight on operations. To the contrary, *Strapi* is significantly affected. When *N* increases, the latency of the related workflows grows rapidly and the throughput decreases instead. This is because the intermediate Web API responses become pretty verbose, bringing in higher parsing overhead. *Gitea* does not show significant differences when *N* is large. This is because the server is already overloaded when *N* is 8. Both `ORFA` and `GraphQL` spend most of their time on *Gitea's* internal operations.

## 4.4 Service Cost

In this section, we compare the peak throughput of `ORFA` with that of `GraphQL` to evaluate the running overhead, or rather service costs. Specifically, we tend to figure out two questions: 1. How much overhead does executing the Wasm program itself brings? 2. How much impact does `ORFA` have on the services when co-located on the same server? For the first question, we defined a synthesized task, which solely commands the server to return a "hello world" string as the result, for both `GraphQL` and `ORFA`. For the second question,

we deployed `ORFA` and `GraphQL` alongside with three applications (in RESTful) on the same machine and collected their throughput metrics. Three three applications maintain the same workloads when co-existing with `ORFA` and `GraphQL`. Also, they only contain read workflows, as the write workflows are not idempotent. The results for both questions are reported in Table 5.

**Table 5: Stress testing results.**

**(a) Detailed results of stress testing with the synthesized workload.**

|  | GraphQL | ORFA-base | ORFA-code | ORFA-inst |
|---|---|---|---|---|
| **TPS** | 909.09 | 222.22 | 384.62 | 1111.11 |
| **Time (ms)** | 1.1 | 4.5 | 2.6 | 0.9 |
| **CPU** | 88% | 100% | 100% | 20% |

**(b) Throughput per second (TPS) results of co-existing `ORFA` with the three realistic applications.**

|  | REST | GraphQL | ORFA-base | ORFA-code | ORFA-inst |
|---|---|---|---|---|---|
| **memos.r** | 260.2 | 97.24 | 77.48 | 120.12 | 130.52 |
| **strapi.r** | 42.1 | 53.34 | 18.9 | 23.058 | 23.352 |
| **gitea.r** | 3.88 | 2.761 | 2.379 | 3.678 | 4.017 |

Table 5a shows that `ORFA` significantly lowers throughput (57% ∼ 76%) and increases latency (134% ∼ 157%) compared to `GraphQL` in the no-caching (`ORFA-base`) and code caching (`ORFA-code`) modes, which can be attributed to the compilation and initialization cost of Wasm programs. On the other hand, when the compilation and initialization are completely eliminated in the instance caching mode (`ORFA-inst`), `ORFA` achieves 22% throughput boost and 18.2% latency reduction with 68% less CPU usage, demonstrating the high performance of Wasm's execution and effectiveness of proposed caching revisions.

Table 5b shows that `ORFA` generally achieves much lower throughput than original RESTful APIs, with a median value of 54.9%, demonstrate the the high cost of using Web API services indirectly through `ORFA`. This is reasonable, as the additional costs not only come from the compilation and initialization of Wasm query programs, but also from the clients' offloaded computation for parsing and assembling HTTP messages. Also, note that `GraphQL` achieves even better result than original RESTful API in *strapi.r*. This falls onto the embedding of `GraphQL` into service code, which eliminates the overhead of a wrapper layer, thereby hinting more potential performance gain by integrating `ORFA` and the service.

## 5 Conclusion

In this paper, we propose `ORFA`, a framework that employs WebAssebmly as a Turing complete query language for Web API services, allowing "One Request For All" operations to eliminate all data round-trips. We present `ORFA`'s programming support and runtime design, explain how to program, run, and debug a Wasm query module. We also introduce two key techniques of module splitting and caching to reduce query module size and query startup latency. Experimental results on representative systems and workloads demonstrate that `ORFA` significantly boosts Web API service efficiency with reduced latency and transmission traffic.

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
