# OpenReview forum: "ORFA: Exploring WebAssembly as a Turing Complete Query Language for Web APIs"
_ACM.org/TheWebConf/2025/Conference — WWW 2025 Oral_

### Official Review · Reviewer_SAau · 2024-11-21

**Novelty:** 5
**Technical Quality:** 4

**Review:**

This paper proposes ORFA (One Request For All), the first in literature that employs WebAssembly (Wasm) as a Web API query language to achieve complete expressiveness of client requests. The experimental evaluation demonstrates that the proposed method outperforms existing approaches in terms of service latency, network traffic robustness, and service cost.

## pros

1. The paper addresses a significant and worthwhile research problem.
2. The authors provide a thorough explanation of the methodology and conduct experimental evaluations.


## cons

1. The testing scenarios fail to fully reflect real-world conditions, limiting the generalizability of the results.
2. The presentation is occasionally unclear, with certain key aspects requiring more precise articulation.

Specific comments are as follows:

## Quality of  Work

This paper is well-structured. It begins by identifying the limitations of REST APIs and GraphQL as query languages and introduces WebAssembly (Wasm) as a proposed solution. The authors argue that Wasm meets strict security requirements while also acknowledging the challenges of using Wasm as a query language. They propose a series of methods to address these challenges.

The authors chose C as the source language for experimental evaluation, primarily due to its simplicity and the maturity of its toolchain. However, this choice does not adequately reflect the performance of all programming languages capable of generating Wasm functions. Other languages might exhibit significantly different, potentially poorer, performance characteristics. This limitation should be acknowledged and discussed to ensure the generality of the claims made in the paper.

The proposed method introduces the challenge of not being able to debug locally. The statement “To enable the recording, clients set the ORFA-Debug header in its requests, and ORFA will respond with the recorded log data instead of the original query result. This way, the query code can be debugged locally like a normal program” is unconvincing. How does providing recorded log data ensure an equivalent experience to traditional local debugging? Moreover, this debugging approach may have inherent limitations, such as reduced interactivity or the inability to replicate complex runtime states fully. The paper should address these concerns and provide a more detailed justification or analysis of the trade-offs involved in this method.

The experimental evaluation lacks completeness. The tests were conducted on queries with relatively low volumes, which do not adequately reflect large-scale, real-world scenarios. In practical applications, the additional compilation and instantiation phases introduced by the proposed method may incur performance overheads absent in prior approaches. Testing on larger datasets is necessary to assess the true performance of the method.

## Clarity

1. In the "Shrinking Query Module Size" section, the discussion on the size of the Wasm program is vague and lacks concrete details.
2. In the experimental evaluation phase, the use of the "OpenAPI-to-GraphQL tool" to generate GraphQL wrappers is described too briefly. The potential impact of this tool on the experiments has not been adequately discussed.

## Originality

The proposed method is novel, offering a general-purpose solution to the limitations of semantically enriched API query languages by eliminating expressive constraints. This approach stands out as a significant contribution to the field.

## Significance

The ORFA method introduced in this paper demonstrates Turing completeness, setting it apart from previous methods. The experimental results highlight its potential performance improvements, suggesting that this method could pave the way for a new paradigm in network query languages.

**Questions:**

1. To what extent does the scale of the experimental data reflect real-world application scenarios?

2. How does the performance overhead introduced by Wasm compilation and instantiation affect the overall performance? Can this overhead be measured explicitly?

**Reviewer Confidence:**

3: The reviewer is confident but not certain that the evaluation is correct

**Scope:**

4: The work is relevant to the Web and to the track, and is of broad interest to the community

---

### Official Review · Reviewer_B3ek · 2024-11-25

**Novelty:** 5
**Technical Quality:** 4

**Review:**

This paper introduces ORFA (One Request For All), a framework that leverages WebAssembly (Wasm) as a Turing-complete query language for Web APIs. ORFA enables complex client-side requests within a single query. ORFA aims to improve API efficiency by minimizing network traffic and reducing latency through module splitting, caching mechanisms, and a Wasm-based runtime environment that securely handles client queries on the server.

Strong points:
1.The paper illustrates positive results.
2.The author proposed some unique approaches to solve the new problem in ORFA.
3.This paper is the first to use Wasm as a query language which is Turing-complete.

**Questions:**

Weakness & Questions:

1.The problem of time consumption on compilation and instantiation is inadequately discussed. It’s better to provide a detailed breakdown of the time consumed in the compilation and instantiation phases.

2.The baseline is insufficiently selected. The selection did not consider other recent API optimization frameworks or query languages. The author may consider Falcor mentioned in related work, or gRPC from google.

3.Readability of this paper needs improving. Certain details and terms lack sufficient explanation(for example, the “__stack_pointer” in 3.3). Moreover, this paper uses too many long sentences, which makes it hard for readers to understand.

**Reviewer Confidence:**

3: The reviewer is confident but not certain that the evaluation is correct

**Scope:**

3: The work is somewhat relevant to the Web and to the track, and is of narrow interest to a sub-community

---

### Official Review · Reviewer_zpe6 · 2024-11-30

**Novelty:** 6
**Technical Quality:** 6

**Review:**

This paper presents a well-structured framework, ORFA, which integrates WebAssembly as a Turing-complete query language for Web APIs. The quality of the work is high because it thoroughly explores the limitations of existing Web API paradigms (e.g., RESTful APIs and GraphQL) and provides practical solutions to address these shortcomings. The paper is also generally clear, with a logical structure and well-defined technical explanations. In particular, the design of ORFA and its components is explained step-by-step, which enhances understanding. This paper demonstrates its originality by proposing a novel approach to applying WebAssembly as a query language for Web APIs and validates the value of this work through its high efficiency, flexibility, and reduced network overhead.

Pros
- A well-written paper with clear and appropriate motivation.
- Comprehensive evaluation metrics, including latency, network traffic, and throughput, using real-world applications.
- The first use of Wasm as a Turing-complete query language in Web API scenarios.
- Demonstrates clear performance improvements over REST and GraphQL, with significant latency reductions.

Cons
- Adopting ORFA may require rewriting existing systems, posing a significant barrier to adoption.
- Limited exploration of potential vulnerabilities associated with adopting a new Wasm-based system (e.g., malicious query modules).
- The low-level nature of Wasm may pose challenges for developers unfamiliar with systems programming or memory management.

**Questions:**

How do you envision the transition process for existing Web API systems (e.g., RESTful or GraphQL) to adopt ORFA? Are there any tools or migration guidelines available to simplify this process?

While ORFA leverages Wasm’s sandboxing for safety, how does it mitigate potential vulnerabilities in dynamic query programs, such as denial-of-service attacks or resource exhaustion caused by malicious queries?

Could you elaborate on how ORFA ensures secure caching, particularly against cache-poisoning attacks or unauthorized query modifications?

In large-scale deployments with high query concurrency, how does ORFA handle caching efficiency and server memory usage? Have you tested the framework under these conditions?

ORFA relies on client applications to generate and transmit Wasm query modules. How do you address concerns regarding Wasm module size and computational resource usage on resource-constrained clients, such as mobile devices? Or are these concerns negligible?

**Reviewer Confidence:**

2: The reviewer is willing to defend the evaluation, but it is likely that the reviewer did not understand parts of the paper

**Scope:**

3: The work is somewhat relevant to the Web and to the track, and is of narrow interest to a sub-community

---

### Official Review · Reviewer_TafT · 2024-12-01

**Novelty:** 3
**Technical Quality:** 3

**Review:**

This work tackles the limitations of current Web API query approaches by introducing ORFA, a novel framework that leverages WebAssembly as a Turing-complete query language. ORFA enables clients to compose complex operations within a single request through WebAssembly's expressiveness, while incorporating specialized module splitting and caching mechanisms to ensure efficient execution. The framework is evaluated on realistic applications, demonstrating significant improvements in both latency and network traffic reduction compared to conventional approaches.

Pros:
S1. ORFA represents an innovative approach by being the first to utilize WebAssembly as a Web API query language, offering complete expressiveness for client requests.
S2. The framework successfully addresses the data under/over-fetching issues common in RESTful APIs through its Turing-complete query capabilities.
S3. The implementation includes practical optimizations like module splitting and caching mechanisms, making it feasible for real-world deployment.

Cons:
W1. The Introduction section lacks a clear articulation of research problems and challenges. Figures 1 and 2 are too abstract and fail to effectively demonstrate the specific challenges addressed in this work (see D1).
W2. The experimental evaluation is limited in scope, as it only compares the proposed method with REST and GraphQL, which provides insufficient validation of the method's effectiveness (see D2).
W3. The paper lacks a rigorous theoretical foundation and formal proofs to support the proposed method (see D3).

**Questions:**

Q1. The research challenges could be better illustrated through more specific and concrete examples in Figures 1 and 2. How could these figures be improved to better demonstrate the unique challenges this work addresses?
Q2. Why were only REST and GraphQL chosen as baselines? Would it be possible to include comparisons with other modern API query approaches to provide more comprehensive validation?
Q3. Could the authors provide more theoretical analysis and proofs to demonstrate why their proposed method is effective? This could include formal proofs of the method's properties and theoretical bounds on its performance.

**Reviewer Confidence:**

3: The reviewer is confident but not certain that the evaluation is correct

**Scope:**

4: The work is relevant to the Web and to the track, and is of broad interest to the community

---

### Official Review · Reviewer_yxsX · 2024-12-02

**Novelty:** 6
**Technical Quality:** 3

**Review:**

I appreciate the authors submitting their work to ACM WebConf. I like the idea of exploring the next frontier of optimizations in Web API communications, but I have some reservations in the design the authors have presented. I think the proposed design poses some risks, with little reward in most cases -- and I wonder when the trade off becomes worth switching from the current state-of-the-art GraphQL.

Pros:
- Novel approach presented: clients can offload computation that involves several network requests to the server, replacing high cost network requests with efficient in-memory requests, which improves performance while minimizing communication cost.

Cons:
- Limited motivation/lack of analysis of feasibility of the approach in real-world applications
- Limited analysis of resource consumption and security and privacy impacts of this approach.
- Limited implementation details of how the experiment was conducted. What was the development effort required to implement the Wasm part?

**Questions:**

Q1. I am not convinced of the motivation for this design. In which real-world use-cases would it be beneficial to use the proposed approach instead of Wasm? For the examples evaluated in Section 4, the number of requests sent by GraphQL don't seem to be much more than Wasm.
Q2. How would this design limit resource consumption at server? In case of limited compute, is it not better to compute on the client? I understand computing on the server in cases where computing on the client would lead to a large number of web requests, but how frequent is that?
Q3. Wasm is not immune to code security issues (https://www.usenix.org/system/files/sec20-lehmann.pdf) -- when executed on the client side, the potential damage radius is limited, but when executed on the server, the radius is significantly larger, do you have any thoughts?
Q4. Wasm was designed for execution in browsers, i.e., JS VMs. Why is Wasm required for such remote code execution? Can any mutually agreed upon (between client and server) platform be used in a sandboxed environment?
Q5. Client-side applications usually operate in a user context, which may contain privacy-sensitive information --  to what extent can the computation be offloaded to the server keeping the payload user context agnostic?
Q6. What is the development effort for developers if this approach were to be adopted? Is it worth the gains? If yes, in which cases? It would be great to have an analysis of that.

**Reviewer Confidence:**

3: The reviewer is confident but not certain that the evaluation is correct

**Scope:**

4: The work is relevant to the Web and to the track, and is of broad interest to the community